# The Incidence and Impact of In-Hospital Bleeding in Patients with Acute Coronary Syndrome during the COVID-19 Pandemic

**DOI:** 10.3390/jcm11102926

**Published:** 2022-05-22

**Authors:** Roberto Licordari, Alessandro Sticchi, Filippo Mancuso, Alessandro Caracciolo, Saverio Muscoli, Fortunato Iacovelli, Rossella Ruggiero, Alessandra Scoccia, Valeria Cammalleri, Marco Pavani, Marco Loffi, Domenico Scordino, Jayme Ferro, Andrea Rognoni, Andrea Buono, Stefano Nava, Stefano Albani, Iginio Colaiori, Filippo Zilio, Marco Borghesi, Valentina Regazzoni, Stefano Benenati, Fabio Pescetelli, Vincenzo De Marzo, Antonia Mannarini, Francesco Spione, Doronzo Baldassarre, Michele De Benedictis, Roberto Bonmassari, Gian Battista Danzi, Mario Galli, Alfonso Ielasi, Giuseppe Musumeci, Fabrizio Tomai, Vincenzo Pasceri, Italo Porto, Giuseppe Patti, Gianluca Campo, Antonio Colombo, Antonio Micari, Francesco Giannini, Francesco Costa

**Affiliations:** 1Department of Clinical and Experimental Medicine, Policlinic “G Martino”, University of Messina, 98124 Messina, Italy; robertolicordari@gmail.com (R.L.); filippo.mancuso09@gmail.com (F.M.); caracciolo.alessandro.ac@gmail.com (A.C.); amicari@unime.it (A.M.); 2GVM Care & Research, Maria Cecilia Hospital, 48033 Cotignola, Italy; sticchialessandro@gmail.com (A.S.); rossellaruggiero1609@gmail.com (R.R.); scoccia.alessandra@gmail.com (A.S.); colombo.emogvm@gmail.com (A.C.); giannini_fra@yahoo.it (F.G.); 3Centro per la Lotta Contro L’Infarto (CLI) Foundation, 00182 Rome, Italy; 4Department of Cardiology, Saint Camillus International University of Health Sciences, 00131 Rome, Italy; 5Department of Cardiovascular Disease, Tor Vergata University, 00133 Rome, Italy; saveriomuscoli@gmail.com (S.M.); v.cammalleri@hotmail.it (V.C.); 6Division of University Cardiology, Cardiothoracic Department, Policlinico University Hospital, 072006 Bari, Italy; fortunato.iacovelli@gmail.com (F.I.); antonia.mannarini@gmail.com (A.M.); francesco.spio@gmail.com (F.S.); 7Cardiovascular Institute, Azienda Ospedaliero-Universitaria di Ferrara, 30010 Cona, Italy; gianlucacampo@unife.it; 8Department of Cardiology, Ospedale Civile SS Annunziata, 12038 Savigliano, Italy; marcopavani@alice.it (M.P.); baldassarre.doronzo@gmail.com (D.B.); micheledebe@gmail.com (M.D.B.); 9Department of Cardiology, Ospedale di Cremona, 26100 Cremona, Italy; loffi.marco@gmail.com (M.L.); valentina.regazzoni@asst-cremona.it (V.R.); gbdanzi@tin.it (G.B.D.); 10Division of Cardiology, Aurelia Hospital, 00165 Rome, Italy; domenicoscordino@hotmail.it (D.S.); fabriziotomai@gmail.com (F.T.); 11U.O.S.D. Cardiologia-Laboratorio di Emodinamica, Dipartimento di Emergenza, Rianimazione e Anestesia, ASST Lariana, Ospedale S. Anna, 22100 Como, Italy; jayme.ferro@gmail.com (J.F.); mario.galli@asst-lariana.it (M.G.); 12Department of Cardiology, Azienda Ospedaliero–Universitaria Maggiore della Carità, 28100 Novara, Italy; andrea.rognoni@maggioreosp.novara.it (A.R.); giuseppe.patti@maggioreosp.novara.it (G.P.); 13Clinical and Interventional Cardiology Unit, Istituto Clinico Sant’Ambrogio, 20149 Milan, Italy; andrebuo@hotmail.com (A.B.); alielasi@hotmail.com (A.I.); 14Division of Invasive Cardiology, ASST Grande Ospedale Metropolitano Niguarda, 20162 Milan, Italy; drstefanonava@gmail.com; 15Department of Cardiology, Mauriziano Hospital, 10128 Turin, Italy; albani.aosta@gmail.com (S.A.); giuseppe.musumeci@gmail.com (G.M.); 16Azienda Unità Sanitaria Locale, IRCCS di Reggio Emilia, 42123 Reggio Emilia, Italy; iginio.colaiori@gmail.com; 17U.O.C. Cardiologia, Ospedale Santa Chiara, 38121 Trento, Italy; filippozi@yahoo.it (F.Z.); borghez@gmail.com (M.B.); roberto.bonmassari@apss.tn.it (R.B.); 18Dipartimento CardioToracoVascolare, Ospedale Policlinico San Martino IRCCS, 16132 Genoa, Italy; stefano.benenati@hotmail.it (S.B.); fabiopescetelli91@gmail.com (F.P.); vincenzodemarzo@gmail.com (V.D.M.); italo.porto@gmail.com (I.P.); 19Department of Cardiology, San Filippo Neri Roma, 00135 Rome, Italy; vpasceri@hotmail.com

**Keywords:** acute coronary syndrome (ACS), COVID-19, myocardial infarction (MI), bleeding, in-hospital outcomes

## Abstract

Background: The COVID-19 pandemic increased the complexity of the clinical management and pharmacological treatment of patients presenting with an Acute Coronary Syndrome (ACS). Aim: to explore the incidence and prognostic impact of in-hospital bleeding in patients presenting with ACS before and during the COVID-19 pandemic. Methods: We evaluated in-hospital Thrombolysis In Myocardial Infarction (TIMI) major and minor bleeding among 2851 patients with ACS from 17 Italian centers during the first wave of the COVID-19 pandemic (i.e., March–April 2020) and in the same period in the previous two years. Results: The incidence of in-hospital TIMI major and minor bleeding was similar before and during the COVID-19 pandemic. TIMI major or minor bleeding was associated with a significant threefold increase in all-cause mortality, with a similar prognostic impact before and during the COVID-19 pandemic. Conclusions: the incidence and clinical impact of in-hospital bleeding in ACS patients was similar before and during the COVID-19 pandemic. We confirmed a significant and sizable negative prognostic impact of in-hospital bleeding in ACS patients.

## 1. Introduction

Patients with acute coronary syndrome (ACS) and concomitant Coronavirus Disease 19 (COVID-19) have an increased risk of morbidity and mortality and often represent a vulnerable population [1]. A number of registries and case reports suggest an increased incidence of thromboembolic events in patients with COVID-19, justified by systemic inflammation, coagulation activation, hypoxemia and immobilization [2,3,4]. A variety of coagulopathies have been reported in association with COVID-19, including disseminated intravascular coagulation (DIC), sepsis-induced coagulopathy (SIC), venous thromboembolism (VTE), arterial thrombotic complications, and thrombo-inflammation [1]. For this reason, anticoagulation at standard prophylactic doses is often considered for all patients admitted with COVID-19, potentially increasing the risk of bleeding [5,6]. Major bleeding is the most common complication in patients admitted for ACS, and its risk is further increased by a combination of multiple antithrombotic agents [7]. The Global Registry of Acute Coronary Events (GRACE) confirmed a 3.9% rate of major bleeding among patients with ACS [8]. In addition, there is a strong relationship between bleeding, mortality, and recurrent myocardial infarction, and prior registries have shown that major bleeding is associated with an up to 60% increase in in-hospital mortality [9,10,11]. The increased complexity of clinical management and the concomitant pharmacological treatment of patients with ACS during the COVID-19 pandemic still represent a therapeutic challenge [12]. The aim of this study was to explore the incidence, characteristics, and prognostic impact of in-hospital bleeding in patients presenting with ACS during the COVID-19 pandemic and to compare them with those of earlier years.

## 2. Materials and Methods

We collected data in a multicenter, national, retrospective ACS collaborative registry from 17 high-volume centers in Italy. The study population consisted of all consecutive patients admitted with ACS during 3 time intervals before (i.e., March–April 2018 and 2019) and during the first wave of the COVID-19 pandemic in the participating centers (i.e., March–April 2020). All patients, regardless of whether they underwent percutaneous coronary intervention, were included. Data about demographics, risk factors, previous medical history, procedural information, and in-hospital outcomes were collected. Acute coronary syndrome diagnosis was made following international guidelines [13,14]. In-hospital major and minor bleeding was evaluated and defined according to the Thrombolysis In Myocardial Infarction (TIMI) definition [15]. Baseline characteristics, procedural characteristics, and clinical outcomes were compared in the two study periods (2018–2019 and 2020). The patients’ privacy was guaranteed by the anonymizing process in the data collection phase. Institutional board approval was obtained by the study promotor institution. The study followed the Strengthening the Reporting of Observational Studies in Epidemiology (STROBE) reporting guidelines for cohort studies.

### Statistical Analysis

Continuous variables have been reported as means and standard deviation (SD) or medians and interquartile ranges, as appropriate. Discrete variables have been indicated as counts and percentages. We compared the two groups of patients with ACS included before and during the COVID-19 pandemic (2018–2019 versus [vs.] 2020) to highlight differences in the demographic, procedural, and clinical characteristics using *t*-test for quantitative variables and chi-square for categorical variables. Univariate and multivariate regression analysis was used to evaluate the association of baseline characteristics with the safety endpoints. Survival analysis using Cox regression was performed to evaluate the prognostic impact of bleeding events in the two study periods, and the heterogeneity of effect was evaluated through interaction testing. Interaction tests were performed with likelihood ratio tests of the null hypothesis that the interaction coefficient was zero. A two-sided probability value < 0.05 was considered significant. All statistical analyses have been computed using SPSS v.26.0 (IBM Corporation, Armonk, NY, USA).

## 3. Results

Among 2851 patients admitted with acute coronary syndrome (ACS), 2142 (75.1%) were admitted in the period of March–April 2018–2019 before the COVID-19 pandemic and 709 (24.9%) during the first wave of the COVID-19 pandemic in March–April 2020. The hospital stay was 6 ± 5.1 days, and the intrahospital mortality rate was 2.4%. Baseline characteristics of patients included before and after the COVID-19 pandemic are provided in Appendix A. The incidence of in-hospital TIMI major or minor bleeding was similar in ACS patients before and during the COVID-19 pandemic (Figure 1). Baseline characteristics for patients suffering or not suffering in-hospital bleeding in the two study periods are reported in Table 1. Patients who suffered in-hospital bleeding and had poorer left ventricle ejection fraction (LVEF) were more likely women, with a history of anemia, chronic kidney disease (CKD), and oncological disease, and with no relevant differences in the two study periods. In-hospital bleeding was more common among patients with procedural complications during coronary intervention, such as hypotension or arrhythmic or mechanical complications. Detailed characteristics of the population admitted during the first wave of the COVID-19 pandemic are reported in Appendix A. During the pandemic, the incidence of bleeding was similar in patients with negative or positive COVID-19 swab (Appendix A), with no relevant differences regarding baseline clinical and procedural characteristics (Appendix A). Univariate and multivariable analysis for TIMI major and minor bleeding is presented in the Appendix A. Kaplan–Meier curves for TIMI major or minor bleeding in patients admitted for ACS before and during the pandemic are presented in Figure 2. Kaplan–Meier curves for patients with positive or negative COVID-19 swab are presented in Appendix A. After adjusting for age and gender, the occurrence of TIMI major or minor bleeding was associated with a significant threefold increase in in-hospital mortality (HR 3.33; 95% CI 1.44–7.69; *p* = 0.005), with no heterogeneity observed for the period before (HR 4.60; 95% CI 1.82–11.64; *p* = 0.001) and during the COVID-19 pandemic (HR 1.20; 95% CI 0.16–8.91; *p* = 0,86) (P_int_ = 0.49) (Figure 3). The results remained consistent when explored by categories of bleeding severity (Appendix A).

## 4. Discussion

The main findings of our study from a large multicenter registry of patients with acute coronary syndrome collected before and during the first wave of the COVID-19 pandemic could be summarized as follows:Traditional baseline predictors of bleeding in ACS patients were similar in the two study periods.The incidence of major or minor bleeding during hospitalization for ACS was similar both before and during the first wave of the COVID-19 pandemic.In-hospital major or minor bleeding carries a significant and immediate prognostic impact in ACS patients, and this was confirmed also in admissions during the COVID-19 pandemic.

To the best of our knowledge this is the first report focusing on the incidence and prognostic impact of in-hospital bleeding in ACS patients during the first wave of the COVID-19 pandemic. With the introduction of refined and more potent antithrombotic regimens, the incidence of ischemic complications in patients with ACS has steadily decreased in the last two decades, while in turn the risk of major bleeding has roughly doubled, becoming the most common clinical complication after an ACS [16].

Bleeding complications have been associated with an increase in mortality in an ACS setting [9,10,11] with major or minor bleeding being associated with a fivefold-higher risk of death during admission [11,17]. Both major bleeding and ischemic events, such as a recurrent MI, have a similar impact on long-term mortality, and depending on bleeding severity, bleeding could have an even higher prognostic impact as high as four times that of a new episode of MI [18].

Among hospitalized patients with COVID-19, thromboembolism is an important piece of the puzzle [1,19]. Autopsy studies demonstrated a high incidence of macro and microthrombi in patients who died after contracting COVID-19 [20]. Leveraging on this observation, several observational studies have suggested a possible benefit for a routine in-hospital use of anticoagulation in this setting [21]. Similarly, the International Society on Thrombosis and Hemostasis (ISTH) and the American Society of Hematology recommend that all hospitalized patients with COVID-19 receive thromboprophylaxis with low molecular weight heparin in order to reduce the risk of thromboembolic complications [5,6,22]. It is not surprising that confirmed or suspected COVID-19 patients are exposed to a more aggressive antithrombotic regimen that could be associated with a higher incidence of intrahospital bleedings. This is especially true in the context of ACS, where multiple antithrombotic therapies with aspirin, a P2Y12 inhibitor, and an initial course of anticoagulation represent the mainstay of treatment alongside coronary revascularization [22]. Based on this assumption, our finding of a similar incidence and prognostic impact of in-hospital bleeding [8] during the first wave of the COVID-19 pandemic could appear counterintuitive. Yet several explanations could be advanced to justify our results.

Multiple national registries confirmed a drastic reduction of roughly 50% of the number of patients presenting with MI compared with the same period in previous years. [12,23] In this context it may be speculated that differences in the baseline characteristics may justify the lack of difference observed in our analysis. Yet baseline characteristics and classic bleeding risk predictors for patients presenting before and during the COVID-19 pandemic appeared similar in our analysis. Also, a significant delay in the clinical presentation of ACS patients during the first wave of the COVID-19 pandemic has been observed in our report and in multiple previous reports. Hence is possible that sicker patients with ACS did not seek assistance or presented late during the COVID-19 pandemic, with a secondary impact on incidence and prognosis of intrahospital bleeding during the pandemic. In fact, not only was an important reduction in ACS presentation rate registered during this period, but it was also seen that the cumulative incidence of out-of-hospital cardiac arrest was strongly associated with the cumulative incidence of COVID-19 [24].

Alternatively, we could speculate that specific adjustment in the multiple antithrombotic therapy in ACS patients with potential indication to additional therapy for confirmed or suspected COVID-19 could have mitigated the risk of bleeding in this observational cohort. In this setting, patients with indication to oral anticoagulation therapy could have been targeted with less potent antiplatelet agents or with shorter courses of multiple antithrombotic therapies, based on clinician risk perception, which could have limited the risk of bleeding complication despite multiple antithrombotic therapy.

## 5. Limitations

Our study has several important limitations that should be acknowledged. First, this is an observational study, hence our findings should be considered hypothesis-generating rather than conclusive. Second, although all patients admitted during the first wave of the COVID-19 pandemic were tested with a specific swab, we included both individuals irrespectively of a confirmed COVID-19 diagnosis or evidence of positive swab. This is relevant, as pharmacologic treatment in this group might have been different. However, our main aim was also to compare predictors of bleeding in patients presenting with ACS both before and during the pandemic. Exploring the frequency and clinical impact of bleeding in a larger population of patients with COVID-19 would be relevant. Importantly, detailed data regarding antithrombotic treatment were not available for all patients included, which limits our ability to explore the relationship between bleeding events and assigned pharmacological therapy. Finally, we only provide outcomes for the in-hospital phase, whereas no information for out-of-hospital events is available. This point is of great importance, and future dedicated studies should target this issue. Nevertheless, this is the first report that focuses on the incidence and prognostic impact of in-hospital bleeding in ACS patients during this unprecedented health crisis, and it could at least partially inform upon the risks of these complications during the in-hospital phase of ACS patients.

## 6. Conclusions

In our large, multicenter registry of ACS patients included before and during the first wave of the COVID-19 pandemic, we observed a similar incidence and prognostic impact of in-hospital bleeding during both study periods. We confirmed a significant and similar negative prognostic impact of in-hospital bleeding during ACS hospitalization, which should prompt physician awareness of this dangerous complication also in the context of COVID-19 when multiple antithrombotic agents may be needed.

## Figures and Tables

**Figure 1 jcm-11-02926-f001:**
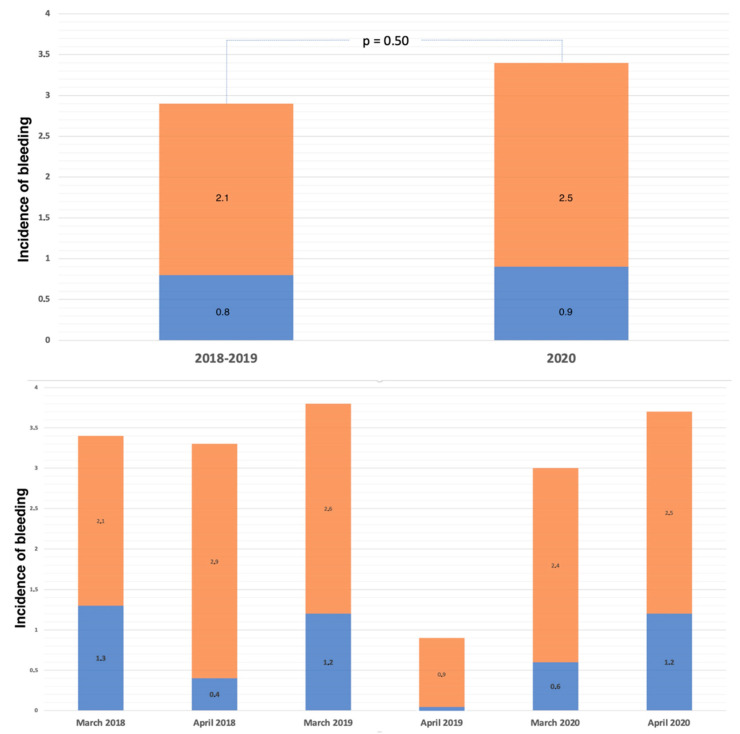
Incidence of in-hospital bleeding before and during the COVID-19 pandemic. Blue segments represent TIMI major bleeding, and orange segments represent TIMI minor bleeding.

**Figure 2 jcm-11-02926-f002:**
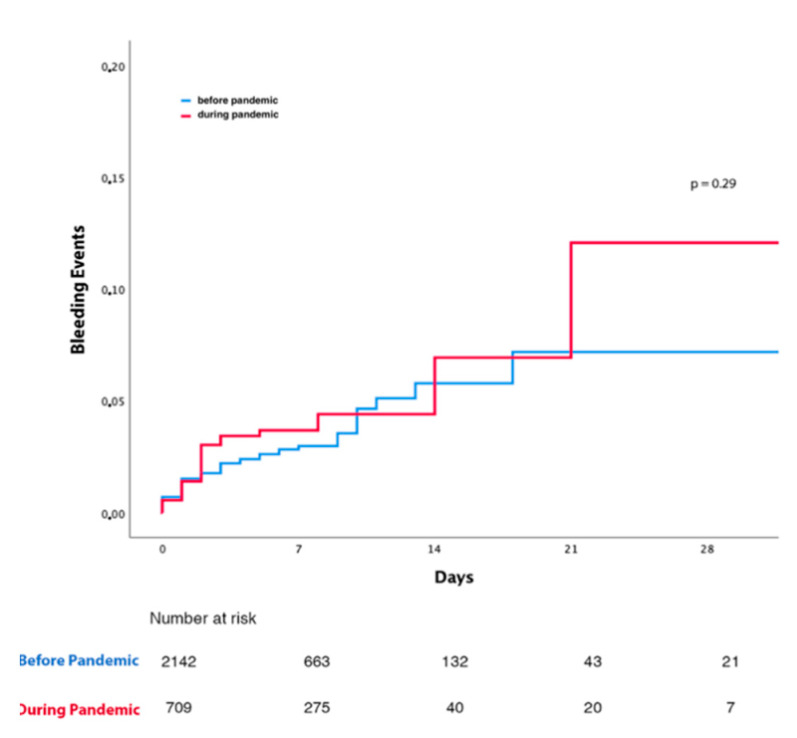
Kaplan-Meier curves for TIMI major or minor bleeding in patients admitted with acute coronary syndrome before and during COVID-19 pandemic.

**Figure 3 jcm-11-02926-f003:**
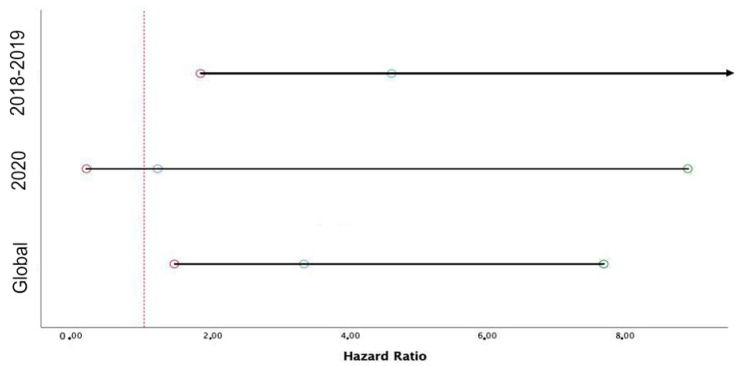
The impact of bleeding on mortality in acute coronary syndrome patients before and during the COVID-19 pandemic.

**Table 1 jcm-11-02926-t001:** Baseline characteristics of the population before and during the pandemic based on the occurrence of in-hospital bleeding.

	Before COVID-19 Pandemic (*n* = 2142)			During COVID-19 Pandemic (*n* = 709)		
	No Bleed (*n* = 2080)	Bleed (*n* = 62)	*p*	No Bleed (*n* = 684)	Bleed (*n* = 25)	*p*
Age	66.9 ± 12.5	69 ± 12.9	0.20	66.8 ± 11.8	69.6 ± 13.22	0.25
BMI (kg/m^2^)	26.4 ± 4.0	26.3 ± 4.1	0.92	26.3 ± 4.0	26.1 ± 3.8	0.87
Female Sex	25.70%	41.70%	0.005	25.50%	37.50%	0.19
Hypertension	69.30%	67.20%	0.73	69.20%	83.30%	0.14
Diabetes Mellitus	27.70%	27.90%	0.97	28.80%	29.20%	0.96
Dyslipidemia	50.60%	37.70%	0.04	50.50%	37.50%	0.21
Smoking	28.00%	24.60%	0.56	30.80%	25.00%	0.54
Ex-smoking	18.10%	23.00%	0.33	14.10%	12.50%	0.83
Atrial Fibrillation (all forms)	9.50%	14.80%	0.17	8.50%	16.70%	0.16
History of Heart Failure	5.10%	6.60%	0.61	4.60%	12.50%	0.08
Valve disease (more than mild)	0.30%	3.30%	<0.001	0.00%	0.00%	N.A.
COPD	7.80%	16.70%	0.01	8.20%	8.30%	0.98
Respiratory/Pulmonary disease	0.50%	1.60%	0.21	0.40%	0.00%	0.74
Neurological disease	0.50%	3.30%	0.004	0.60%	0.00%	0.70
Chronic kidney disease (GFR < 60 mL/min)	11.20%	21.30%	0.01	12.80%	25.00%	0.08
Hemorrhagic diathesis	1.00%	4.90%	0.004	0.00%	0.00%	N.A.
Thrombotic diathesis	1.10%	3.30%	0.12	0.40%	0.00%	0.74
Anemia	0.30%	4.90%	<0.001	0.30%	0.00%	0.79
Inflammatory/Infective disease	1.80%	1.60%	0.93	1.20%	0.00%	0.59
Previous oncological disease	1.20%	6.60%	0.001	0.40%	4.20%	0.01
Previous PCI	22.60%	24.60%	0.71	22.00%	16.70%	0.53
Previous CABG	5.70%	3.30%	0.42	4.60%	12.50%	0.07
Previous MI	20.30%	24.60%	0.41	19.20%	16.70%	0.75
Previous Stroke/TIA	4.10%	11.50%	0.006	4.70%	0.00%	0.27
Atypical symptoms at presentation	13.00%	24.60%	0.009	14.80%	16.70%	0.80
Dyspnea	12.50%	31.10%	<0.001	13.30%	16.70%	0.63
Respiratory impairment	5.70%	24.60%	<0.001	6.90%	8.30%	0.78
Fever	1.60%	3.30%	0.32	4.90%	8.30%	0.46
Heart Failure (at the presentation)	12.80%	21.30%	0.05	12.50%	12.50%	0.99
Killip >1 at presentation	29.4%	44.8%	0.01	39.8%	29.2%	0.29
Night Presentation	22.10%	27.90%	0.28	20.40%	12.50%	0.34
EF (%; at presentation)	48.7 ± 9.8	45.3 ± 10	0.009	47.3 ± 9.9	42 ± 9.0	0.02
Time Door to Balloon (minutes)	315 ± 2504	274 ± 573	0.92	228 ± 573	138 ± 322	0.50
Time Symptoms to Cath-lab door (minutes)	1043 ± 3339	1108 ± 2718	0.90	1264 ± 3689	651 ± 975	0.45
Time Symptoms to Emergency call (minutes)	533 ± 1607	612 ± 2322	0.77	817 ± 2670	828 ± 1848	0.98
Cardiac arrest before cathlab	3.50%	6.60%	0.20	2.80%	8.30%	0.11
STEMI	43.2%	443%	0.16	46.9%	56	0.49
NSTEMI	39.20%	42.60%	0.16	35.80%	36.00%	0.49
Unstable Angina	13.10%	4.90%	0.16	9.60%	8.00%	0.49
MINOCA	3.2%	5.2%	0.20	4.2%	0%	0.23
TakoTsubo Syndrome	1.3%	3%	0.75	3.5%	0%	0.29
Thrombotic occlusion	37.60%	31.10%	0.30	37.30%	47.80%	0.30
Thrombus Aspiration	17.50%	9.80%	0.12	14.70%	8.30%	0.38
Number of stent implanted	0.82 ± 0.34	0.79 ± 0.41	0.53	0.80 ± 0.45	0.80 ± 0.50	0.96
Fibrinolysis	0.20%	1.60%	0.04	0.00%	0.00%	N.A.
GP IIB/IIIA use	10.90%	14.80%	0.35	9.70%	0.00%	0.11
Any Ventricular Support	3.00%	6.50%	0.12	3.20%	8.00%	0.19
Arrhythmic complications during procedure	4.30%	8.30%	0.14	4.30%	12.50%	0.05
Intrahospital Arrhythmic Complications	2.90%	11.70%	<0.001	2.50%	16.70%	<0.001
Mechanical Complications	1.10%	8.30%	<0.001	2.40%	12.50%	0.003

BMI = body mass index; COPD = chronic obstructive pulmonary disease; PCI = percutaneous coronary intervention; CABG = coronary artery bypass graft; MI = myocardial infarction; TIA = transient ischemic attack; EF = ejection fraction; STEMI = ST-segment elevated myocardial infarction; NSTEMI = non ST-segment elevated myocardial infarction; GP = glycoprotein; MINOCA = Myocardial Infarction with Non Obstructive Coronary Arteries; N.A. = Not Available.

## Data Availability

Not applicable.

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
