# Peer review of "The Incidence and Impact of In-Hospital Bleeding in Patients with Acute Coronary Syndrome during the COVID-19 Pandemic"

_jcm, 2022, doi:10.3390/jcm11102926_

Round 1
Reviewer 1 Report
The manuscript entitled “Incidence and Impact of In-Hospital Bleeding in Patients with Acute Coronary Syndrome during the COVID-19 Pandemic” was reviewed. This study attempted to explore an incidence and a prognostic impact of in-hospital bleeding in patients presenting with acute coronary syndrome (ACS) before and during the COVID-19 pandemic. This manuscript is an interesting and well written. However, there are some critical issues that should be properly addressed.
- Although readers can calculate the number of patients with and without bleeding from Figure 1, the authors should describe the number of patients with and without bleeding in the Table. (i.e. No Bleed; n=XXX, Bleed; n=XXX, etc.)
- The reviewer thinks the Table of patient characteristics simply comparing “before COVID-19 wave” with “during COVID-19 wave” should be provided.
- Please show the details of antithrombotic therapy (both of anticoagulant and antiplatelet agents) during hospitalization, if possible.
- As the authors described, some patients with ACS might be deceased before arrival at the hospital in COVID-19 wave. However, the reviewer guesses there were some sicker patients with delayed arrival. How many patients arrived at the hospital in a serious condition (Killip>1) before and during COVID-19 wave, respectively? Further, what is the reason that non-bleed group had more Killip>1 patient than bleed group during COVID-19 wave?
- Please briefly describe the reason that major and minor bleeding were not influence on the mortality in the patients during COVID-19 wave.
- Please provide the information about PCI success rate (final TIMI flow grade) and details of major bleeding, if possible.
Author Response
Reviewer 1
This study attempted to explore an incidence and a prognostic impact of in-hospital bleeding in patients presenting with acute coronary syndrome (ACS) before and during the COVID-19 pandemic. This manuscript is an interesting and well written. However, there are some critical issues that should be properly addressed.
Answer: Many thanks for the appreciation of our work.
- Although readers can calculate the number of patients with and without bleeding from Figure 1, the authors should describe the number of patients with and without bleeding in the Table. (i.e. No Bleed; n=XXX, Bleed; n=XXX, etc.)
Answer: Thank you. We added the number of patients with and without bleeding in Table 1.
- The reviewer thinks the Table of patient characteristics simply comparing “before COVID-19 wave” with “during COVID-19 wave” should be provided.
Answer: Thank you for raising this important point. We have now provided a new Supplementary Table 1 comparing patient characteristics before and during COVID-19 pandemic.
We also commented this novel table in the Results section, page 3, lines 101-103 as follows: “Baseline characteristics of patients included before and during COVID-19 pandemics is provided in Supplementary Table 1”
- Please show the details of antithrombotic therapy (both of anticoagulant and antiplatelet agents) during hospitalization, if possible.
Answer: Thank you for addressing this important point. We agree with the Reviewer that a precise description of the assigned antithrombotic treatment would have been informative. Unfortunately detailed data regarding assigned antithrombotic treatment in the two study periods have not been collected in our multicenter collaborative registry. We recognize this limits our ability to strengthen the correlation between observed bleeding events and the assigned antithrombotic treatment, despite we would expect a standard approach based on international guideline recommendation as most of the centers included in the registry are tertiary level institutions that routinely manage ACS patients.
Yet, to highlight this potential limitation in our analysis, we have pointed out in limitation section of the manuscript (page 7, lines 207-208) the following statement “Importantly, detailed data regarding antithrombotic treatment, were not available for all patients included, which limits our ability to explore the relationship between bleeding events and assigned pharmacological therapy.”
- As the authors described, some patients with ACS might be deceased before arrival at the hospital in COVID-19 wave. However, the reviewer guesses there were some sicker patients with delayed arrival. How many patients arrived at the hospital in a serious condition (Killip>1) before and during COVID-19 wave, respectively? Further, what is the reason that non-bleed group had more Killip>1 patient than bleed group during COVID-19 wave?
Answer: We agree with Reviewer 1 on this point. As we stated in the paper and in line with other reports in the literature, ACS patients during the COVID-19 pandemic first wave arrived later in the hospital and in general with higher grade of hemodynamic instability. In fact, as presented in the new Supplementary Table 1, also in our population we observed higher rates of Killip>1 at presentation and need for ventricular support during first wave COVID-19 compared to the period before. Nevertheless, irrespective of the higher clinical complexity, the rates of in-hospital bleeding and their clinical consequences in ACS patients were similar in the two study periods. This highlights the fact that in-hospital bleeding risk is not only bound to hemodynamic stability at presentation but to multiple other factors that instead were similar before and after COVID-19. In fact, relevant predictors of in-hospital bleeding such as age, renal function, weight and sex were similar in the two study periods.
With respect to the second point raised by Reviewer 1, rates of Killip>1 patients among non-bleed patients during the COVID-19 first-wave was not statistically significant (p=0.29) and the nominal difference present might, in our opinion, be due to a play of chance.
- Please briefly describe the reason that major and minor bleeding were not influence on the mortality in the patients during COVID-19 wave.
Answer: We thank again for raising this important question. The smaller sample size in the group of patients presenting during the COVID-19 pandemics, and the wider confidence interval may justify our inability to observe a statistically significant impact on mortality in this subgroup. Yet, in our opinion, a more correct interpretation of this subgroup analysis of patients presenting before and during COVID-19 pandemic, should be based on the analysis of the interaction testing. In our multivariable analysis we observed that in-hospital bleeding was associated with a 3 fold increased risk of in-hospital mortality, with no significant interaction between the two explored study periods and the impact of bleeding on mortality. This gives support to the hypothesis that in-hospital bleeding is associated with a higher mortality risk both before and during COVID-19 first-wave.
- Please provide the information about PCI success rate (final TIMI flow grade) and details of major bleeding, if possible.
Answer: Unfortunately, PCI final TIMI flow grade was not within the focus of this analysis and was not collected.
Reviewer 2 Report
The present study is aimed as assessing the incidence, characteristics and prognostic impact of in-hospital bleeding in patients presenting with ACS during the COVID 19 pandemic and compare it with those of earlier years. The purpose and objective of this analysis is interesting. There are several major issues to be considered:
- Authors state that they compared the two groups of patients with ACS included before and during the COVID-19 pandemic. However, I was unable to see this comparison. Table 1 compares patients with and without bleeding during each period. I think that, before this comparison, global characteristics of populations before and during COVID-19 should be compared to properly assess potential differences between both time-periods. This could be performed in global and breaking down by bleeding event. In the same manner, a comparison between the 639 patients negative COVID swab and the 70 patients positive COVID swab should also be performed.
- It should be important to know where the collected information comes from. Is it a dedicated ACS registry?...was the information extracted from administrative databases?... is there a possibility of missing patients?....if so, is there a possibility of inclusion bias?...
- It is important to highlight that you are analyzing global differences between two time-periods, which likely does not allow to make conclusions about the specific population with ACS and positive COVID swab. In this sense your analysis is more focused on the assessment of the health care system for patients with ACS before and within COVID pandemic. On the other hand, it is surprising that only 70 patients admitted for ACS in 17 centers during the first wave tested positive for COVID. Although authors already indirectly pointed out this in the limitations section, it would be important to justify that there is not a risk of substantial inclusion bias (i.e. many patients with ACS are not detected as COVID positive).
- Authors stated that no heterogeneity was observed for the period before and during pandemic regarding the association between bleeding event and mortality. However, although is true that the statistical interaction was negative, it surprises that the magnitude of the differences was relevant (HR 4.60 vs 1.20 before and during pandemic respectively), and the absence of interaction seems to be due to the imprecise estimation of the association between bleeding and mortality during pandemic (CI 0.16-8.91), which is consequence of the low bleeding rate. So, in this sense, the study could be considered underpowered to detect potential differences.
- Statistical analysis. Apparently it was adjusted for age and gender. Why?...should other variables be considered to adjust for?
- In the whole manuscript there is not mention on the absolute number of patients with bleeding event. It should be amended.
Author Response
Reviewer 2
The present study is aimed as assessing the incidence, characteristics and prognostic impact of in-hospital bleeding in patients presenting with ACS during the COVID 19 pandemic and compare it with those of earlier years. The purpose and objective of this analysis is interesting. There are several major issues to be considered:
Authors state that they compared the two groups of patients with ACS included before and during the COVID-19 pandemic. However, I was unable to see this comparison. Table 1 compares patients with and without bleeding during each period. I think that, before this comparison, global characteristics of populations before and during COVID-19 should be compared to properly assess potential differences between both time-periods. This could be performed in global and breaking down by bleeding event. In the same manner, a comparison between the 639 patients negative COVID swab and the 70 patients positive COVID swab should also be performed.
Answer: We would like to thank Reviewer 2 for her/his thorough evaluation. In line with Reviewer 2 comment, and as also suggested by Reviewer 1, we now provide a new Supplementary Table 1 that highlights the comparison of the two study periods and a Supplementary Table 2 with the comparison between patients with positive and negative swab.
We also commented this novel table in the Results section, page 3, lines 101-103 as follows: “Baseline characteristics of patients included before and during COVID-19 pandemics is provided in Supplementary Table 1” and in lines 111-112 “Detailed characteristics of the population admitted during the COVID-19 pandemic first wave are reported in Supplementary Table 2.”
It should be important to know where the collected information comes from. Is it a dedicated ACS registry?...was the information extracted from administrative databases?... is there a possibility of missing patients?....if so, is there a possibility of inclusion bias?...
Answer: We again thank Reviewer 2 for raising this important point. Data were collected retrospectively among 17 centers in a collaborative dedicated ACS registry. All consecutive patients presented during the evaluated study periods (March-April 2018, 2019, 2020) from the 17 contributing centers were included with no exceptions. Hence, we believe that the risk of inclusion bias is small. We have now better explained for clarity this point in the methods section as follows: “We collected data in a multicenter, national, retrospective ACS collaborative registry, from 17 high-volume centers in Italy. The study population consisted of all consecutive patients admitted with ACS during 3 time-intervals before (i.e.e March/April 2018 and 2019) and during the first wave of the COVID-19 pandemic in the participating centers (i.e. March/April 2020).”
It is important to highlight that you are analyzing global differences between two time-periods, which likely does not allow to make conclusions about the specific population with ACS and positive COVID swab. In this sense your analysis is more focused on the assessment of the health care system for patients with ACS before and within COVID pandemic. On the other hand, it is surprising that only 70 patients admitted for ACS in 17 centers during the first wave tested positive for COVID. Although authors already indirectly pointed out this in the limitations section, it would be important to justify that there is not a risk of substantial inclusion bias (i.e. many patients with ACS are not detected as COVID positive).
Answer: We again thank Reviewer 2 for this interesting comment. We agree with Reviewer 2 that our analysis is exploring results on a health system perspective, but we remain convinced that the analysis of the results of epidemiologic data during this unprecedented medical crisis is of great relevance also to understand the social and logistical consequences of the pandemic. Unfortunately, the relatively low number of patients included prevent us from providing definitive conclusions, but still give us the possibility of raising potentially relevant hypotheses. We agree with Reviewer 2 that the total number of patients testing positive was small, but at the same time, we underscore that we included only patients that presented with an ACS during a relatively short time-frame (March-April 2018-2019-2020) matching with COVID-19 first-wave in Italy. With respect to the concern of inclusion bias raised by Reviewer 2, we highlight again that all consecutive patients from the 17 contributing institutions were included in the registry, hence we remain convinced that there is a very small probability of introducing bias. In addition, during the first-wave of the pandemic all patients admitted for ACS were tested for COVID19, so again we believe that inclusion bias is unlikely.
In order to highlight the important point raised by Reviewer 2 we have now added in the limitations section (page 7, lines 202-204) the following statements: “Second, although all patients admitted during the first wave of the COVID-19 pandemic were tested with a specific swab, we included both individuals irrespectively of a confirmed COVID-19 diagnosis or evidence of positive swab.” and “Exploring the frequency and clinical impact of bleeding in a larger population of patients with COVID-19 would be relevant”
Authors stated that no heterogeneity was observed for the period before and during pandemic regarding the association between bleeding event and mortality. However, although is true that the statistical interaction was negative, it surprises that the magnitude of the differences was relevant (HR 4.60 vs 1.20 before and during pandemic respectively), and the absence of interaction seems to be due to the imprecise estimation of the association between bleeding and mortality during pandemic (CI 0.16-8.91), which is consequence of the low bleeding rate. So, in this sense, the study could be considered underpowered to detect potential differences.
Answer: We again thank the Reviewer for the outstanding comment. We tested interaction between the two time-periods adjusted for possible confounders. Although, we agree with Reviewer 2 that owing to the low event rate the wider confidence intervals, especially during the COVID-19 first wave could reduce our ability of derive firm conclusions.
With the current data analysis we were notable to observe a significant prognostic impact of in-hospital bleeding in the two time periods. If anything, it could have been expected that, owing to the higher complexity of the management of patients during the pandemic, a worse prognosis of bleeding events during this period to be present, but we were not able to see such an increased in risk. In fact, we did not observe neither an increased rate of in-hospital bleeding complication in ACS patients in the two period, neither a worse prognostic impact, which supports that in-hospital bleeding is multifactorial and not only bound to patient hemodynamic instability.
In order to highlight the possibility that the lack of difference between the subgroups is due to an intrinsic lack of power, we have better highlighted this point in the limitation. However, this is the largest registry that explored the clinical impact of bleeding before and during the pandemic first wave and we remain convinced that our data could be interesting and the clinical findings are sound.
Statistical analysis. Apparently it was adjusted for age and gender. Why?...should other variables be considered to adjust for?
Answer: Unfortunately, we were not able to adjust for many other covariates owing to the relatively small number of events to prevent model overfitting. Age and gender are two strong and recognized risk factors for in-hospital bleeding and we decided to use these to adjust the model exploring the clinical impact of bleeding.
In the whole manuscript there is not mention on the absolute number of patients with bleeding event. It should be amended.
Answer: Thanks. This comment was also raised by Reviewer 1. We have now added the number of patients with and without bleeding in Table 1.
Round 2
Reviewer 2 Report
No additional comments on my side.